# The Differential Systemic Biological Effects between Computer Navigation and Conventional Total Knee Arthroplasty (TKA) Surgeries: A Prospective Study

**DOI:** 10.3390/jpm12111835

**Published:** 2022-11-03

**Authors:** Shu-Jui Kuo, Ka-Kit Siu, Kuan-Ting Wu, Jih-Yang Ko, Feng-Sheng Wang

**Affiliations:** 1School of Medicine, China Medical University, Taichung 404, Taiwan; 2Department of Orthopedic Surgery, China Medical University Hospital, Taichung 404, Taiwan; 3Department of Orthopedic Surgery, Park One International Hospital, Kaohsiung 813, Taiwan; 4Department of Orthopedic Surgery, Kaohsiung Chang Gung Memorial Hospital, Kaohsiung 833, Taiwan; 5Department of Medical Research, Kaohsiung Chang Gung Memorial Hospital, Kaohsiung 833, Taiwan

**Keywords:** total knee arthroplasty (TKA), navigation TKA, soluble P-selectin, matrix metalloproteinase-9 (MMP-9), C-reactive protein (CRP), interleu-kin-8 (IL-8)

## Abstract

Distal femur reaming-free total knee arthroplasty (TKA) was reported to possess lower risk of acute myocardial infarction (AMI) or venous thromboembolism (VTE) than conventional TKA in a retrospective population-based study. We tried to offer prospective biological evidence by comparing the levels of AMI and VTE serum surrogate markers among the patients undertaking navigation and conventional TKAs to support these observations. Thirty-four participants undertaking navigation TKA and 34 patients receiving conventional TKA were recruited between February 2013 and December 2015. Blood samples were drawn from all participants before TKA, and 24 and 72 h after TKA, to assess the concentration of soluble P-selectin, matrix metalloproteinase-9 (MMP-9), C-reactive protein (CRP), and interleukin-8 (IL-8) between the participants undergoing navigation and conventional TKAs. We showed that significantly lower serum levels of soluble P-selectin 24 h after, as well as CRP 24 and 72 h after TKA could be observed in the navigation cohort. The more prominent surge of serum soluble P-selectin and CRP were perceived 24 and 72 h after TKA among the participants undergoing conventional TKA. Based upon our prospective biological evidence, the merits of navigation TKA are strengthened by lower levels of AMI and VTE serum surrogate markers.

## 1. Introduction

Total knee arthroplasty (TKA) is an acknowledged procedure for the treatment of advanced knee osteoarthritis and could bring about high satisfaction [1]. However, TKA is not free from complications, and these complications could be potentially fatal [2,3]. The absolute 6-week risk of acute myocardial infarction (AMI) was 0.21% among TKA patients [2]. Venous thromboembolism (VTE) after TKA can manifest from a subclinical manifestation of deep vein thrombosis (DVT) to a potentially deadly pulmonary embolism (PE). The incidence of in-hospital VTE after TKA is about 0.9%~1.9% [3]. Thus, it is an important issue to mitigate the risk of AMI and/or VTE after TKA.

P-selectin belongs to the selectin family and is bound on the membranes of the Weibel–Palade bodies of endothelial cells and α-granules of platelets, and soluble P-selectin is derived from the alternative splicing [4]. Soluble P-selectin not only reflects the extent of platelet activation but also directly induces a coagulation cascade [5]. Matrix metalloproteinase-9 (MMP-9), also known as gelatinase B, is associated with both arterial atherosclerotic plaque instability and vein wall biomechanics during thrombus formation [6,7]. C-reactive protein (CRP) directly takes part in the arterial atherosclerotic process and is a known predictor of novo ruptured plaque and vascular death [8]. CRP also affects VTE risk by inducing monocytes to express tissue factor [9]. Interleukin-8 (IL-8), a C-X-C chemokine, mediates the activation of integrin-mediated adhesion of neutrophils and triggers the adhesion of monocytes to vascular endothelium [10,11]. Elevated plasma IL-8 levels are related to an increased risk of coronary artery disease and VTE in population-based studies [10,11,12]. P-selectin, MMP-9, CRP, and IL-8 are all involved in the processes of both arterial and venous hemostatic processes.

Prosthetic malalignment is an important contributing factor for painful TKA prosthesis [13]. The navigation-assisted or robotic technique was initially employed to optimize the prosthetic alignment. However, under the navigation-assisted or robotic TKA, bone cutting can be executed precisely in the extra-medullary way, thus diminishing the violation of the bone marrow cavity. Diminishing bone marrow insult is an additional benefit other than the optimization of prosthetic alignment. Some authors have indicated the relationships between the violation of bone marrow cavity and the risk of AMI and VTE after TKA [14,15]. A recent population-based retrospective study suggested that the distal femur reaming-free TKA seems to have lower risk of AMI and/or VTE than the conventional TKA. However, support of these observations from prospectively designed clinical studies and biological corroboration are lacking [16].

Due to the unresolved issue, we conducted a prospective study to offer biological evidence for the observations derived from a recent retrospective population-based study. We hypothesize that the mitigation of the intramedullary reaming of the bone marrow of the distal femur might alleviate the postoperative increases of selected markers involved in both arterial and venous hemostatic processes. This study prospectively compared the selected markers between the patients undertaking navigation and conventional TKAs as the surrogate contrast of AMI and VTE. We hypothesize that differential levels of markers involved in both arterial and venous hemostatic processes could be observed after navigation and conventional TKAs.

## 2. Materials and Methods

Our study was approved by the Institutional Review Board of Chang Gung Memorial Hospital (IRB number: 100-0038A3, 101-0050C, 104-9154D) and registered in ClnicalTrials.gov website (registration number: NCT02206321). Written informed consents were obtained from all the participants. The patients undergoing TKA surgeries performed by the senior author (J.-Y.K.) and professor Ching-Jen Wang between February 2013 and December 2015 were all enrolled. The participants were self-separated in accordance with their clinical registration to the two senior authors. Professor Ching-Jen Wang performed conventional TKA, and the senior author (J.-Y.K.) performed navigation TKA. Both surgeons had performed more than one thousand conventional or navigation TKAs, respectively. Elderly (>90 years), those with malignancy, autoimmune disease, undergoing renal replacement therapy, active infection, previous AMI or VTE history, any kind of surgery performed less than 6 months before the index TKA, and any previous surgery on the knee receiving index TKA (such as fracture repair operation or high tibia osteotomy) were excluded from the study. All the enrolled patients submitted eligible informed consent and knew their group assignment before index TKAs, as the patients undertaking navigation TKAs had four additional tiny stab wounds for tracker-pinning.

Navigation TKAs were carried out under the guidance of the navigation system (Vector Vision, Brain Lab, Heimstetten, Germany, CT-free). Both tibia and femur bone cutting were carried out in an extramedullary way among patients assigned to the navigation group, and femoral bone cuts were led by intramedullary instruments among patients allocated to the conventional group.

Ten milliliters of venous blood attained from each participant before TKA and 24 and 72 h after surgery were collected and stored at −80 °C until analysis. Concentrations of soluble P-selectin, MMP-9, CRP, and IL-8 were quantified using ELISA kits (R & D Systems Inc., Minneapolis, MN, USA). Each patient started continuous passive range of motion exercises and quadriceps strengthening exercises 24 h after TKA, and off-bed ambulation was suggested for most patients on the same day. Oral aspirin 500 mg was given every day for VTE prophylaxis if not contraindicated. Doppler ultrasonography or venography would be arranged if DVT was clinically suspicious (e.g., positive Homan’s sign, increased calf or thigh circumference, etc.), whereas, lung perfusion scan or chest CT were arranged if PE was suspected. For the patients free from DVT suspicion, rigorous physiotherapy and off-bed ambulation was arranged for the mechanical prophylaxis of VTE.

Hemoglobin (Hb) concentrations were obtained prior to and 24 h after TKA. If the patients had Hb < 8 mg/dL or Hb 8~9 mg/dL with unstable vitals, blood transfusion with packed red blood cells was ordered. Oral iron supplement was given for patients with Hb 8~9 mg/dL with stable vital signs. Clinical evaluations for DVT were performed regularly until discharge and at regular outpatient follow-ups.

The least sample size necessary for the two groups was pre-determined by the G*Power 3.1.9.2 software (http://www.gpower.hhu.de/en.html (accessed on 1 January 2013)) before the recruitment of participants. The priori power calculation employed a 2-tailed Wilcoxon signed rank test to reckon the sample size of at least 27 for each group (α level: 0.05; power: 80%; effect size: 0.8; allocation ratio: 1).

The data were shown as median (lower quartile, upper quartile). Categorical variables were compared by the Chi-square testing or Fisher’s exact test as appropriate. The Mann–Whitney U test was applied to compare the between-group differences. The Friedman test was utilized for the assessment of repeated within-group measurements for continuous variables, and the Wilcoxon signed-rank test was employed for post hoc analysis. All statistics were carried out by IBM SPSS software version 21.0 (IBM Corp. Armonk, NY, USA), and the *p* value of < 0.05 was defined to be statistically significant.

## 3. Results

After excluding one patient undergoing conventional surgery without offering valid informed consent, sixty-eight eligible patients offering valid informed consent between February 2013 and December 2015 were recruited. There were five males and twenty-nine females in the navigation group (67.0 (64.0, 73.0) years old). The conventional group comprised 10 males and 24 females (66.0 (61.0, 69.0) years old). There were no differences in gender (*p* = 0.144), age (*p* = 0.300), operated side (*p* = 0.215), type of preoperative coronal plane deformity (*p* = 0.230), or body mass index (*p* = 0.124) between the two groups. There were no between-group differences in the proportion of subjects suffering from diabetes (*p* = 0.355), hypertension (*p* = 1.000), coronary artery disease (*p* = 1.000), and stroke (*p* = 1.000) (Table 1).

Before TKA, serum concentrations were commensurate between the two groups for soluble P-selectin (*p* = 0.897), MMP-9 (*p* = 0.682), CRP (*p* = 0.263), and IL-8 (*p* = 0.992). The *p*-values for the Friedman test were 0.013, <0.001, <0.001 and <0.001 for soluble P-selectin, MMP-9, CRP, and IL-8 in the navigation group; corresponding values were <0.001, <0.001, <0.001 and <0.001 in the conventional group. The respective *p*-values that were less than 0.050 of the post-hoc Wilcoxon signed-rank test are provided in Table 2, Table 3, Table 4 and Table 5.

At 24 h after TKA, the median serum soluble P-selectin (37.46 (27.85; 62.82)) in the navigation group was 45.2 % (*p* < 0.001) lower than that in the conventional group (68.35 (45.63; 70.97)). The median increment in serum soluble P-selectin was 46.7% (21.94 (−0.79; 36.17)) and 44.7% (19.50 (−1.19; 43.71)) lower in the navigation cohort versus that in the conventional group (41.20 (18.57; 55.63)) and (43.59 (17.55; 53.44)) 24 h (*p* = 0.002) and 72 h (*p* = 0.046) after TKA, respectively (Table 2) (unit: ng/mL).

The median serum CRP (5782.79 (5530.72; 6516.73)) was 23.1% lower in the navigation group than that in the conventional cohort (7522.12 (7009.85; 8018.18)) (*p* < 0.001) 24 h after TKA. At 72 h after surgery, the median serum CRP 5987.70 (5804.76; 7429.90) in the navigation group was 26.3% (*p* < 0.001) lower than that in the conventional group (8127.07 (7630.61; 8161.80)). The median increment in serum CRP was 32.2% lower (*p* = 0.016) in the navigation group (4260.38 (3373.61; 5334.24)) versus that in the conventional group (6280.45 (3053.30; 6780.72)) 24 h after TKA. At 72 h after operation, the median increment in serum CRP was 40.1% lower (*p* = 0.004) in the navigation cohort (4098.27 (2700.19; 5332.74)) versus that in the conventional group (6843.20 (5039.93; 7513.91)) (Table 4) (Unit: ng/mL).

There were no between-group differences before and 24, 74 h after TKA in terms of MMP-9 and IL-8 (Table 2 and Table 5).

According to a previous population-based study, AMI risk peaked to 30.9-fold during the first 2 weeks but did not differ from controls thereafter among TKA patients [2]. The median time to the presentation of DVT and PE was 20 and 12 days after TKA, respectively, and the cumulative risk of VTE lasted for one month after TKA [17]. Based upon these previous studies, all the participants among the two cohorts were followed for 6 weeks after index TKA surgeries. None of the participants suffered from AMI or symptomatic VTE within 6 weeks after TKA.

## 4. Discussion

The applications of technology-assisted (e.g., navigation, robotics) TKA surgeries have recently been gaining popularity. One recent US nationwide study tried to identify elective primary TKA patients from 2015 to 2020 [16]. Among 847,496 patients enrolled, 49,317 (5.82%) and 24,460 (2.89%) underwent computer navigation (CN)-TKA and robotic assistance (RA)-TKA, respectively. CN-TKA application and RA-TKA deployment increased from 5.64% to 6.41% and 0.84% to 5.89%, respectively, from 2015 to 2020. After adjusting for confounders, CN-TKA was related to lower pulmonary embolism (*p* < 0.001), acute respiratory failure (*p* = 0.015), and periprosthetic joint infection (*p* = 0.001) risk compared to conventional TKA. RA-TKA was associated with lower myocardial infarction (*p* = 0.013), pulmonary embolism (*p* = 0.001), and deep vein thrombosis (*p* < 0.001) risk than conventional TKA. Lower opioid prescription was observed with CN-TKA and RA-TKA than with conventional TKA (*p* < 0.001) 24 h after operation. Lower postoperative opioid consumption was also seen in RA-TKA (*p* < 0.001) 24 h after surgery. In keeping with these findings, our group also showed that the risk of periprosthetic joint infection is lower in the navigation TKA cohort than in the conventional TKA cohort [18]. These data reinforce the safety of RA-TKA and CN-TKA compared to conventional TKA.

In addition to the acute complications requiring admission, the differential systemic biological effects between conventional and technology-assisted TKAs have become an issue attracting increasing attention. The previous works from us and other groups trying to demonstrate the differential levels of systemic biological factors are summarized in Table 6. The present study offered novel prospective biological evidence to support the observations that the risks of AMI and VTE are lower among patients undertaking distal femur reaming-free TKAs in a recent retrospective population-based study [16].

In our study, we tried to suggest the potential risk reduction effect for arterial and venous thromboembolic events for navigation TKAs by comparing serum soluble P-selectin, MMP-9, CRP, and IL-8 between the two TKA techniques. We demonstrated that significantly lower serum levels of soluble P-selectin 24 h after, as well as CRP 24 and 72 h after TKA could be observed in the navigation cohort. The more prominent surge of serum soluble P-selectin and CRP were perceived 24 and 72 h after TKA among the participants undergoing conventional TKA. The differential expression of serum soluble P-selectin and CRP after navigation and conventional TKA deserves attention.

The association between serum level of soluble P-selectin and the occurrence of AMI and VTE has been reported before. The concentration of plasma-soluble P-selectin has been shown to be positively correlated with the Gensini score, the numbers of vessels lesions, and the type of coronary artery lesion among patients with angiocardiography-documented coronary heart disease [23]. Ramacciotti et al. has shown that a combination of serum-soluble P-selectin level and Wells score could establish (cut point ≥ 90 ng/mL + Wells ≥ 2) (specificity: 96%; positive predictive value: 100%) and refute DVT diagnosis (cut point ≤ 60 ng/mL + Wells < 2) (sensitivity: 99%; specificity: 33%; negative predictive value: 96%) [24]. Interestingly, the median postoperative serum-soluble P-selectin levels were below and above 60 ng/mL for the navigation and conventional cohort, respectively.

Serum CRP levels also correlate with the onset of both arterial and venous thromboembolic effects as well. According to one Taiwanese study, the serum CRP levels were remarkably higher in subjects with AMI onset < 6 h than in individuals with stable angina (2.7 ± 2.3 mg/L vs. 1.4 ± 0.7 mg/L, *p* < 0.001) and in control individuals (2.7 ± 2.3 mg/L vs. 1.0 ± 0.6 mg/L, *p* < 0.001) [8]. Kunutsor et al. established a prospective study with 25-year follow up and concurrent meta-analysis to unravel the correlation between CRP level and VTE risk. In this 25-year longitudinal study, CRP was metered in blood samples at baseline from 2420 men. After age adjustment for 119 VTE patients, the regression dilution ratio for log CRP was 0.57 (95% CI: 0.51–0.64). The meta-analysis part of the study merged nine studies comprising 2225 VTE cases and 81625 participants. The fully adjusted estimated VTE risk was 1.14 (95% CI: 1.08–1.19) per standard deviation increase in log baseline CRP. The pooled risk estimate for VTE per 5 mg/L increase in CRP levels was 1.23 (95% CI: 1.09–1.38). The author thus concluded that increased CRP is related to greater VTE risk [25]. The publications mentioned above demonstrated the potential predictive capacity of soluble P-selectin and CRP for AMI and VTE events. The findings of our study cannot be extrapolated to assert that navigation TKAs lead to fewer post-TKA AMI and VTE. The relationships between the serum VTE markers and the true occurrence of AMI VTE among patients undergoing TKA necessitate a larger cohort for further validation.

Previous studies have shown that MMP-9 activity and expression level are increased during thrombus resolution in vivo [26]. MMP-9 mediates macrophage migration and macrophage-induced thrombus resolution [27,28]. Increased level of IL-8 is favorable for thrombus resolution, and impaired inflammatory response by cytokine receptor deletion or neutropenia also decreases thrombus resolution [28,29,30]. In our study, no between-group differences of serum MMP-9 concentration could be observed. The decoupling of the expression of thrombus-formation marker (soluble P-selectin) and thrombus-resolution marker (MMP-9, IL-8) between the navigation and conventional cohort might reflect the increased net thrombus accumulation in the conventional cohort. The phenomenon of decoupled thrombus formation and resolution and the differential VTE events between the two groups warrant further investigation.

Our study is not flawless. Firstly, we could not prospectively show the superiority of navigation over conventional TKA regarding the true AMI and VTE incidences because of the small sample size. Secondly, the patients were free to choose which senior surgeon they favored, thus jeopardizing the randomization process, in spite of the rigorous anonymization and de-identification process. Thirdly, we performed clinical physical assessment to exclude DVT only in this study. More detailed clinical assessment by the application of modern functional tools, such as baropodometry, should be performed for the participants undertaking TKA [31]. Despite these limitations, the participants did not know which technique would be performed before admission, and no participant shifted to the other group before the surgery. Baseline demographic profiles were comparable between the two groups. The strategies mentioned above might lessen the potential influence of selection bias.

## 5. Conclusions

Conventional TKAs contribute to higher gush of serum AMI and VTE surrogate markers. Our study demonstrates lower systemic AMI and VTE surrogate markers as emerging biochemical signatures that complement the known advantages of navigation TKAs. We offered novel prospective biological evidence to support the observed lower risk of AMI and VTE for distal femur reaming-free TKAs.

## Figures and Tables

**Table 1 jpm-12-01835-t001:** The demographic profiles between the navigation and conventional groups.

	Navigation (*n* = 34)	Conventional (*n* = 34)	*p*-Value
Gender (male/female)	5/29	10/24	0.144
Age (years)	67.0 (64.0; 73.0)	66.0 (61.0; 69.0)	0.300
Side (left/right)	16/18	11/23	0.215
Deformity (valgus/varus)	9/25	5/29	0.230
Body mass index (kg/m^2^)	27.10 (25.70; 29.20)	28.70 (25.70; 31.00)	0.124
Diabetes	8	5	0.355
Hypertension	23	23	1.000
Coronary artery disease	1	2	1.000
Stroke	1	0	1.000

**Table 2 jpm-12-01835-t002:** Serum soluble P-selectin concentrations (ng/mL) before TKA, then at 24 and 72 h after TKA.

	Navigation (*n* = 34) ^A^	Conventional (*n* = 34) ^B^	# *p*-Value
Baseline	20.36 (9.55; 41.83) ^ab^	15.19 (11.54; 35.86) ^cd^	0.897
24 h	37.46 (27.85; 62.82) ^a^	68.35 (45.63; 70.97) ^c^	< 0.001
72 h	43.15 (29.25; 66.93^) b^	62.26 (33.4;, 66.47) ^d^	0.352
24 h-baseline	21.94 (−0.79; 36.17)	41.20 (18.57; 55.63)	0.002
72 h-baseline	19.50 (−1.19; 43.71)	43.59 (17.55; 53.44)	0.046

# Mann-Whitney U test, ^A^: *p* = 0.013; ^B^: *p* < 0.001 by Friedman test, ^a^: *p* = 0.015; ^b^: *p* = 0.003; ^c^: *p* < 0.001; ^d^: *p* < 0.001 by post-hoc Wilcoxon signed-rank test.

**Table 3 jpm-12-01835-t003:** Serum metalloproteinase-9 (MMP-9) concentrations (ng/mL) before TKA, then at 24 and 72 h after TKA.

	Navigation (*n* = 34) ^A^	Conventional (*n* = 34) ^B^	# *p*-Value
Baseline	243.14 (167.04; 369.10) ^ab^	217.56 (172.23; 354.21) ^cd^	0.682
24 h	841.79 (583.57; 1073.78) ^a^	903.34 (505.68; 1310.89) ^c^	0.818
72 h	927.37 (681.93; 1430.69) ^b^	1119.43 (594.61; 1846.24) ^d^	0.920
24 h-baseline	520.88 (353.43; 771.57)	576.94 (279.84; 1094.34)	0.803
72 h-baseline	724.08 (423.66; 1054.85)	710.52 (357.28; 1563.52)	0.904

# Mann–Whitney U test, ^A^: *p* < 0.001; ^B^: *p* < 0.001 by Friedman test, ^a^: *p* < 0.001; ^b^: *p* < 0.001; ^c^: *p* < 0.001; ^d^: *p* < 0.001 by post-hoc Wilcoxon signed-rank test.

**Table 4 jpm-12-01835-t004:** Serum C-reactive protein concentrations (ng/mL) before TKA, then at 24 and 72 h after TKA.

	Navigation (*n* = 34) ^A^	Conventional (*n* = 34) ^B^	# *p*-Value
Baseline	1282.17 (350.45; 2741.22) ^ab^	1059.40 (681.85; 4560.36) ^cd^	0.263
24 h	5782.79 (5530.72; 6516.73) ^a^	7522.12 (7009.85; 8018.18) ^c^	<0.001
72 h	5987.70 (5804.76; 7429.90) ^b^	8127.07 (7630.61; 8161.80) ^d^	<0.001
24 h-baseline	4260.38 (3373.61; 5334.24)	6280.45 (3053.30; 6780.72)	0.016
72 h-baseline	4098.27 (2700.19; 5332.74)	6843.20 (5039.93; 7513.91)	0.004

# Mann–Whitney U test. ^A^: *p* < 0.001; ^B^: *p* < 0.001 by Friedman test. ^a^: *p* < 0.001; ^b^: *p* < 0.001; ^c^: *p* < 0.001; ^d^: *p* < 0.001 by post-hoc Wilcoxon signed-rank test.

**Table 5 jpm-12-01835-t005:** Serum interleukin-8 concentrations (pg/mL) before TKA, then at 24 and 72 h after TKA.

	Navigation (*n* = 34) ^A^	Conventional (*n* = 34) ^B^	# *p*-Value
Baseline	13.48 (9.80; 28.46) ^ab^	15.08 (10.19; 32.20) ^cd^	0.992
24 h	61.49 (29.23; 65.79) ^a^	62.94 (32.06; 70.33) ^c^	0.263
72 h	58.30 (30.35; 66.96) ^b^	38.88 (33.20; 65.48) ^d^	0.881
24 h-baseline	27.39 (7.93; 52.12)	28.61 (12.31; 55.20)	0.617
72 h-baseline	32.90 (2.55; 50.41)	21.45 (0.74; 52.93)	0.826

# Mann–Whitney U test, ^A^: *p* < 0.001; ^B^: *p* < 0.001 by Friedman test, ^a^: *p* < 0.001; ^b^: *p* < 0.001; ^c^: *p* < 0.001; ^d^: *p* < 0.001 by post-hoc Wilcoxon signed-rank test.

**Table 6 jpm-12-01835-t006:** The comparison of systemic biological factors between the conventional and the technique avoiding intramedullary reaming.

	Participants	Design	Results
Kuo et al. (2015) [19]	44N/33C	Prospective cohort	Patients undergoing navigation TKAs had fewer blood loss and lower CAMs in serum and hemovac drainage. Milder post-op elevation of serum PECAM-1 (*p* = 0.003) and ICAM-1 (*p* = 0.022) from the pre-op basis was also noted.
Kuo et al. (2018) [1]	44N/53C	Prospective cohort	Serum levels of IL-6, IL-10, TNF-α and TGF-β1 were increased from baseline by smaller increments in the navigation cohort compared with the conventional cohort at 24 h and at 72 h. IL-10 levels in hemovac drainage 24 h after TKA were also significantly lower in the navigation cohort.
Siu et al. (2019) [20]	89N/85C	Prospective cohort	A decreased plasma D-dimer level and a less apparent increase in the plasma D-dimer level were seen in patients undertaking navigation TKA in contrast to patients undertaking conventional TKA 24 h after operation.
Kayani et al. (2021) [21]	30R/30C	Prospective RCT	Participants undertaking conventional TKA and robotic TKA had commensurate changes in the systemic inflammatory and localized thermal response at 6 h, 24 h, 48 h, and day 28 after surgery. Robotic TKA had apparently lower levels of IL-6 (*p* < 0.001), TNF-α (*p* = 0.021), ESR (*p* = 0.001), CRP (*p* = 0.004), lactate dehydrogenase (*p* = 0.007), and creatine kinase (*p* = 0.004) at day 7 after operation compared with conventional TKA.
Xu et al. (2022) [22]	34R/31C	Retrospective	IL-6 serum concentrations were apparently lower in the MA-TKA group on postop day 1 (11.4 (5.2, 21.0) vs. 24.6 (86.3, 170.8), *p* = 0.031). This difference in inflammatory markers was more pronounced at 3 days after the TKA because IL-6, ESR, CRP and CK values were significantly lower in the MA-TKA group 72 h after operation (*p* < 0.05).

N: navigation; C: conventional; R: robotic; CAM: cell adhesion molecule; ICAM: intercellular cell adhesion molecule; VCAM: vascular cell adhesion molecule; PECAM: platelet endothelial cellular adhesion molecule-1; IL-6: interleukin-6; IL-12: interleutkin-10; TNF-α: tumor necrosis factor-alpha; TGF-β1: transforming growth factor-beta 1; IL-10: interleukin-10; ESR: erythrocyte sedimentation rate; CRP: C-reactive protein; MA-TKA: MAKO-robotic assisted total knee arthroplasty; CK: creatine kinase.

## Data Availability

The data presented in this study are available on request from the corresponding author.

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
