# Peer review of "The Differential Systemic Biological Effects between Computer Navigation and Conventional Total Knee Arthroplasty (TKA) Surgeries: A Prospective Study"

_jpm, 2022, doi:10.3390/jpm12111835_

Round 1

Reviewer 1 Report (New Reviewer)

This paper looks at the biological markers for AMI and VTE comparing conventional TKA methods and computer navigation. Essentially looking at whether distal femur intramedullary reaming to insert intramedullary distal femur cutting guide increases these biological markers.

This is just a lab analysis but does not look at whether these patients actually had clinical evidence of VTE or patients suffered from MI.

The Introduction, Methods, Statistics, Results and Discussion is written well.

There are a few spelling/grammatical mistakes that can be corrected in the revision process.

The paper is well written and I would recommend publication of this paper.

It is of great value to orthopaedics surgeons especially those justifying the use of robots and navigation.

Maybe include a short statement/outcome of these patients in terms of AMI and VTE.

Thank you

Author Response

Thank you wholeheartedly for rigorously reviewing our work and offering constructive and informative advice.

Indeed, whether these patients actually had clinical evidence of VTE and AMI, not only the lab analysis, are important. We thus add a short statement/outcome of these patients in terms of AMI and VTE: “According to previous population-based study, AMI risk peaked to 30.9-fold during the first 2 weeks but did not differ from controls thereafter among TKA patients[2]. The median time to the presentation of DVT and PE was 20 and 12 days after TKA respectively, and the cumulative risk of VTE lasted for one month after TKA[17]. Based upon these previous studies, all the participants among the two cohorts were followed for 6 weeks after index TKA surgeries. None of the participants suffered from AMI or symptomatic VTE within 6 weeks after TKA.” (line 171~177).

Spelling/grammatical mistakes (e.g.: “arrnged” (line 103), TNF-a and TGF-b1 (Table 6 captions), “with regard to true AMI” revised to “regarding the true” (line 253)) have been corrected.

Thanks again for your kind instruction !

Reviewer 2 Report (New Reviewer)

Dear Authors, the topic is estremely interesting in fact the relationship between navigated tka and conventional tka in terms of severe complications  is very important

As regards the introduction i suggest to improve this section by describing the topic in general In fact it  is important to underline the relationship between conventional tka (intramedullarry alignment) and navigated tka in terms of mechanical alignment. For this reason i suggest to cite the following article in which  Authors speak about the alignment and the painful TKA. 

-Painful knee prosthesis: CT scan to assess patellar angle and implant malrotation

Spinarelli A. et al.

Muscles, Ligaments and Tendons JournalOpen Access Volume 6, Issue 4, Pages 461 - 4661 October 2016

As regards the M&M, the section is well described but it is important to underline that notwithstanding the modern functional evalution such as baropodometry, Author performed an evaluation of leg in orders to exclude the complications of TVP only.

For this reason i suggest to cite the following article:

-Baropodometry on patients after total knee arthroplasty

Notarnicola A. et al.

Musculoskeletal SurgeryVolume 102, Issue 2, Pages 129 - 1371 August 2018

As regards the discussion and conclusions, are balanced and well supported by analysis . 

Author Response

Thank you wholeheartedly for rigorously reviewing our work and offering constructive and informative advice.

As for the improvement of introduction section, “Prosthetic malalignment is an important contributing factor for the painful TKA prosthesis[13]. The navigation assisted or robotic technique was initially employed to optimize the prosthetic alignment. However, under the navigation assisted or robotic TKA, bone cutting can be executed precisely in the extramedullary way, thus diminishing the violation of the bone marrow cavity. Diminishing bone marrow insult is an additional benefit other than the optimization of prosthetic alignment.” Has been added following your instruction. Professor Spinarelli A.’s work has been cited as reference 13.

As for the issue of clinical assessment and application of baropodometry, the following two paragraphs have been revised: “Doppler ultrasonography or venography would be arranged if DVT was clinically suspicious (e.g., positive Homan’s sign, increased calf or thigh circumference…), whereas lung perfusion scan or chest CT were arranged if PE was suspected.” (line 101~103);  “Thirdly, we performed clinical physical assessment to exclude DVT only in this study. More detailed clinical assessment by the application of modern functional tool, such as baropodometry, should be warranted for the participants undertaking TKA” (line 256~259). Professor Notarnicola A’s work has been cited as reference 31.

We are thankful for your introducing two informative paper for us, and we have learned a lot during the revision process. Thanks again for your kind instruction !

This manuscript is a resubmission of an earlier submission. The following is a list of the peer review reports and author responses from that submission.

Round 1

Reviewer 1 Report

Dear Editor, Thank you for your kind invitation to review this manuscript. 

The subject of this manuscript, cannot be accepted for publication in its current version to fulfill the quality requirements for publication, and The manuscript needs to be revised to obtain a more clear text.

However 

in its current form, the manuscript is confusing, the novelty is not clear, the English language is confusing.

- The Abstract should contain the purpose of the research, a very short methodology and the most important conclusions.

The conclusion is not clear should be rewritten!